# 1 Structural optimisation of wind turbine towers based on finite

2

# element analysis and genetic algorithm

Lin Wang<sup>1\*</sup>, Athanasios Kolios<sup>1</sup>, Maria Martinez Luengo<sup>1</sup>, Xiongwei Liu<sup>2</sup>

<sup>1</sup>Centre for Offshore Renewable Energy Engineering, School of Water, Energy and Environment, Cranfield University, Cranfield, MK43 0AL, UK

<sup>2</sup>Entrust Microgrid, Lancaster Environment Centre, Gordon Manley Building, Lancaster University,

LA1 4YQ, UK

## 3 Abstract

### 4

5 A wind turbine tower supports the main components of the wind turbine (e.g. rotor, nacelle, drive train components, etc.). The structural properties of the tower (such as stiffness and natural frequency) can 6 7 significantly affect the performance of the wind turbine, and the cost of the tower is a considerable portion of the overall wind turbine cost. Therefore, an optimal structural design of the tower, which has a 8 9 minimum cost and meets all design criteria (such as stiffness and strength requirements), is crucial to 10 ensure efficient, safe and economic design of the whole wind turbine system. In this work, a structural 11 optimisation model for wind turbine towers has been developed based on a combined parametric FEA (finite element analysis) and GA (genetic algorithm) model. The top diameter, bottom diameter and 12 13 thickness distributions of the tower are taken as design variables. The optimisation model minimises the 14 tower mass with six constraint conditions, i.e. deformation, ultimate stress, fatigue, buckling, vibration and 15 design variable constraints. After validation, the model has been applied to the structural optimisation of a 16 5MW wind turbine tower. The results demonstrate that the proposed structural optimisation model is capable of accurately and effectively achieving an optimal structural design of wind turbine towers, which 17 18 significantly improves the efficiency of structural optimisation of wind turbine towers. The developed 19 framework is generic in nature and can be employed for a series of related problems, when advanced 20 numerical models are required to predict structural responses and to optimise the structure.

21

# 22 **1. Introduction**

23

24 Wind power is capable of providing a competitive solution to battle the global climate change and energy 25 crisis, making it the most promising renewable energy resource. As an abundant and inexhaustible energy resource, wind power is available and deployable in many regions of the world. Therefore, regions such as 26 Northern Europe and China are making considerable efforts in exploring wind power resources. According 27 to Global Wind Energy Council (GWEC, 2016), the global wind power cumulative capacity reached 432 28 29 GW at the end of 2015, growing by 62.7 GW over the previous year. It is predicted that wind power could reach a total installed global capacity of 2,000 GW by 2030, supplying around 19% of global electricity 30 31 (Council, 2015).

<sup>\*</sup> Corresponding author. Tel.: +44(0)1234754706; E-mail address: lin.wang@cranfield.ac.uk

A wind turbine tower supports the main components of the wind turbine (e.g. rotor, nacelle, drive train 33 components, etc.) and elevates the rotating blades at a certain elevation to obtain desirable wind characteristics. The structural properties of a wind turbine tower, such as the tower stiffness and natural 34 35 frequency, can significantly affect the performance and structural response of the wind turbine, providing adequate strength to support induced loads and avoiding resonance. Additionally, the cost of the tower is a 36 significant portion of the overall wind turbine cost (Aso and Cheung, 2015). Therefore, an optimal 37 structural design of the tower, which has a minimum cost and meets all design criteria (such as stiffness 38 39 and strength requirements), is crucial to ensure efficient, safe and economic design of the whole wind 40 turbine system. It also contributes to reducing the cost of energy, which is one of the long-term research 41 challenges in wind energy (van Kuik et al., 2016).

The structural optimisation model of a wind turbine tower generally consists of two components, i.e. 1) a 44 wind turbine tower structural model, which analyses the structural performance of the tower, such as tower 45 mass and deformations; and 2) an optimisation algorithm, which deals with design variables and searches 46 for optimal solutions.

Structural models used for wind turbine towers can be roughly classified into two groups, i.e. 1D (one-48 49 dimensional) beam model and 3D (three-dimensional) FEA (finite element analysis) model. The 1D beam 50 model discretises the tower into a series of beam elements, which are characterised by cross-sectional 51 properties (such as mass per unit length and cross-sectional stiffness). Due to its efficiency and reasonable 52 accuracy, the 1D beam model has been widely used for structural modelling of wind turbine towers (Zhao 53 and Maisser, 2006, Murtagh et al., 2004) and blades (Wang et al., 2014b, Wang et al., 2014a, Wang, 2015). 54 Although it is efficient, the beam model is incapable of providing some important information for the 55 tower design, such as detailed stress distributions within the tower structure, hence making such models incapable of capturing localised phenomena such as fatigue. In order to obtain the detailed information, it 56 57 is necessary to construct the tower structure using 3D FEA. In 3D FEA, wind turbine towers are generally constructed using 3D shell or brick elements. Compared to the 1D beam model, the 3D FEA model 58 59 provides more accurate results and is capable of examining detailed stress distributions within the tower structure. Due to its high fedility, the 3D FEA model has been widely used for modelling wind turbine 60 structures (Wang et al., 2015, Wang et al., 2016b, Stavridou et al., 2015). Therefore, the 3D FEA model is 61 62 chosen in this study to model the wind turbine tower structure.

Optimisation algorithms can be roughly categorised into three groups (Herbert-Acero et al., 2014), i.e. exact algorithms, heuristic algorithms and metaheuristic algorithms. Exact algorithms, which find the best solution by evaluating every possible combination of design variables, are very precise because all possible combinations are evaluated. However, they become time-consuming and even infeasible when the number of design variables is large, requiring huge computational resources to evaluate all possible combinations. Heuristic algorithms, which find near-optimal solutions based on semi-empirical rules, are more efficient

than exact algorithms. However, they are problem-dependent and their accuracy highly depends on the 71 accuracy of semi-empirical rules, limiting their applications to some extent. Metaheuristic algorithms, 72 which are more complex and intelligent heuristics, are high-level problem-independent algorithms to find 73 near-optimal solutions. They are more efficient than common heuristic algorithms and are commonly 74 based on optimisation processes observed in the nature, such as PSO (particle swarm optimisation) 75 (Kennedy, 2011), SA (simulated annealing) (Dowsland and Thompson, 2012) and GA (genetic algorithm) 76 (Sivanandam and Deepa, 2007). Among these metaheuristic algorithms, the GA, which searches for the 77 optimal solution using techniques inspired by genetics and natural evolution, is capable of handling a large 78 number of design variables and avoiding being trapped in local optima, making it the most widely used 79 metaheuristic algorithm (Wang et al., 2016a). Therefore, the GA is selected in this study to handle the 80 design variables and to find the optimal solution.

This paper attempts to combine FEA and GA to develop a structural optimisation model for onshore wind turbine towers. A parametric FEA model of wind turbine towers is developed and validated, and then coupled with GA to develop a structural optimisation model. The structural optimisation model is applied to a 5MW onshore wind turbine to optimise the 80m-height tower structure.

This paper is structured as follows. Section 2 presents the parametric FEA model of wind turbine towers.
Section 3 presents the GA model. Section 4 presents the optimisation model by combining the parametric
FEA model and GA model. Results and discussions are provided in Section 5, followed by conclusions in
Section 6.

## 92 2. Parametric finite element analysis (FEA) model of wind turbine towers

## 94 2.1. Model description

A parametric FEA model of wind turbine towers is established using ANSYS, which is a widely used 97 commercial FE software. The parametric FEA model enables the design parameters of wind turbine towers 98 to be easily modified to create various tower models. The flowchart of the parametric model of wind 99 turbine towers is presented in Fig. 1.

- 101
- Figure 1. Flowchart of the parametric FEA model for wind turbine towers
- Each step of the flowchart Fig. 1 is detailed below.

1) Define design parameters: In the first step, design parameters of the wind turbine towers, such as tower

top and bottom diameters, are defined.

2) Create tower geometry: The tower geometry is created based on the bottom-up approach, which creates

- low dimensional entities (such as lines) first and then creates higher dimensional entities (such as areas) on 109 top of low dimensional entities.
- 3) Define and assign material properties: In this step, material properties (such as Young's modulus and
- Poisson's ratio) are defined and then assigned to the tower structure.
- 4) Define element type and generate mesh: Due to the fact that wind turbine towers are generally thin-wall structures, they can be effectively and accurately modelled using shell elements. The element type used here is the shell element Shell281, which has eight nodes with six degrees of freedom at each node and it is well-suited for linear, large rotation, and/or large strain nonlinear applications. Additionally, a regular quadrilateral mesh generation method is used to generate high quality element, ensuring the computational accuracy and saving on computational time.
- 5) Define boundary conditions: In this step, boundary conditions are applied. The types of boundary
  conditions are dependent on the types of analyses. For instance, a fixed boundary condition is applied to
  the tower bottom for modal analysis.
- 6) Solve and post-process: Having defined design parameters, geometry, materials, element types, mesh and boundary conditions, a variety of analyses (such as static analysis, modal analysis and buckling analysis) can be performed. The simulation results, such as tower deformations and stress distributions, are then plotted using post-processing functions of ANSYS software.

#### 126 **2.2. Validation of the parametric FEA model**

- A case study is performed to validate the parametric FEA model of wind turbine towers. The NREL 5MW
- wind turbine (Jonkman et al., 2009), which is a representative of large-scale of HAWTs is chosen as an

example. The NREL 5MW wind turbine is a reference wind turbine designed by NREL (National Renewable Energy Laboratory), and it is a conventional three-bladed upwind HAWT, utilising variablespeed variable-pitch control. The geometric and material properties of NREL 5MW wind turbine tower are presented in Table 1. The steel density is increased from a typical value of 7,850 kg/m<sup>3</sup> to a value of 8,500 kg/m<sup>3</sup> to take account of paint, bolts, welds and flanges that are not accounted for in the tower thickness data (Jonkman et al., 2009). The diameters and thickness of the tower are linearly tapered from the tower base to tower top.

Table 1. Geometric and material properties of the NREL 5MW wind turbine tower (Jonkman et al., 2009)

| Properties                    | Values |
|-------------------------------|--------|
| Tower height [m]              | 87.6   |
| Tower top outer diameter [m]  | 3.87   |
| Tower top wall thickness      | 0.0247 |
| Tower base outer diameter [m] | 6      |
| Tower base wall thickness [m] | 0.0351 |
| Density [kg/m <sup>3</sup> ]  | 8500   |
| Young's modulus [GPa]         | 210    |
| Shear modulus [GPa]           | 80.8   |

The parametric FEA model presented in Section 2.1 is applied to the modal analysis of the NREL 5MW wind turbine tower. In this case, the tower is fixed at the tower bottom and free-vibration (no loads on the 141 tower), and tower head mass is ignored. A regular quadrilateral mesh generation method is used to generate 142 143 high quality elements. In order to determine the appropriate mesh size, a mesh sensitivity study is carried 144 out for the first 6 modal frequencies, of which the results are presented in Table 2. As can be seen from Table 2, the modal frequencies converge at a mesh size of 0.5m, with a maximum relative difference 145 (0.002%) occurring for the 2<sup>nd</sup> side-to-side mode when compared to further mesh refinement with a mesh 146 size of 0.25m. Therefore, 0.5m is deemed as the appropriate element size. The created mesh is presented in 147 148 Fig. 2, and the total number of element is 6,960.

150

| Table | 2. | FEA    | mesh | sensitivity | analysis |
|-------|----|--------|------|-------------|----------|
| 14010 |    | 1 12/1 | meon | Sensitivity | unuiyon  |

| Modal frequencies       | 2m sizing | 1m sizing | 0.5m sizing | 0.25m sizing |
|-------------------------|-----------|-----------|-------------|--------------|
| 1 <sup>st</sup> SS (Hz) | 0.8781    | 0.8782    | 0.8782      | 0.8782       |
| 1 <sup>st</sup> FA (Hz) | 0.8855    | 0.8855    | 0.8856      | 0.8856       |
| 2 <sup>nd</sup> SS (Hz) | 4.2315    | 4.2305    | 4.2276      | 4.2275       |
| $2^{nd}$ FA (Hz)        | 4.2463    | 4.2469    | 4.2429      | 4.2428       |

(where SS refers to side-to-side; FA refers to force-aft)

| 1 | 5 | 2 |
|---|---|---|
| 1 | 5 | 3 |

Figure 2. Mesh of NREL 5MW wind turbine tower

Table 3 compare the results from the present FEA model against the results from ADAMS software reported in Ref. (Jonkman and Bir, 2010).

Table 3. Mode frequencies of NREL 5MW wind turbine tower

| Mode frequencies        | ADAMS (Jonkman | Present FEA model | %Diff |
|-------------------------|----------------|-------------------|-------|
|                         | and Bir, 2010) |                   |       |
| 1 <sup>st</sup> SS (Hz) | 0.8904         | 0.8782            | 1.37  |
| 1 <sup>st</sup> FA (Hz) | 0.8904         | 0.8856            | 0.54  |
| 2 <sup>nd</sup> SS (Hz) | 4.3437         | 4.2276            | 2.67  |
| $2^{nd}$ FA (Hz)        | 4.3435         | 4.2429            | 2.32  |

As can be seen from Table 3, the force-aft (FA) and side-to-side (SS) tower modal frequencies calculated from the present FEA model match well with the results reported in Ref. (Jonkman and Bir, 2010), with the maximum percentage difference (2.67%) occurring for the  $2^{nd}$  SS mode. This confirms the validity of the present parametric FEA model of wind turbine towers.

## 165 **2.3. Application of parametric FEA model to a 5MW wind turbine tower**

The parametric FEA model is applied to FEA modelling of a 5MW wind turbine tower. The geometry and 168 material properties, mesh, boundary conditions used in the FEA modelling are presented below.

169

### 170 2.3.1. Geometry and material properties

171

The geometric and material properties of 5MW wind turbine tower are presented in Table 4. Again, the steel density is increased from a typical value of 7,850 kg/m<sup>3</sup> to a value of 8,500 kg/m<sup>3</sup>, taking account of paint, bolts, welds and flanges that are not accounted for in the tower thickness data. The tower height is 80m, and other geometric information (i.e. tower top diameter, tower bottom diameter and tower thickness

<sup>159</sup>