# Peer review of "Structural optimisation of wind turbine towers based on finite"

_Wind Energy Science, 2016_

## Referee Comment (RC1) · Anonymous Referee #1 · 18 Jan 2017

General comment:

The manuscript addresses sizing optimization of onshore wind turbine towers by combining finite element analyses performed in commercial software with genetic algorithm. Tower top diameter, tower bottom diameter, and thickness distributions are used as design variables in the minimization of mass, constrained by variable bounds, deformation constraints, ultimate stress constraints, fatigue constraints, buckling constraints, and vibration constraints.

The topic is worth investigating and important for further improving the design of wind turbine towers, and the topic is suited for Wind Energy Science. However, I do have some concerns about the modeling of the tower which are listed in the specific comments section.

[Figure]

The introduction is lacking a more thorough overview of structural optimization of wind turbine towers/support structures, which is the main aspect of the paper. It seems that mainly optimization using GA is listed, and these references appear in a later section (Section 3). Also, most references in the paper are to blade design rather than tower design.

Specific comments:

Line 64-65 The categorization of optimization algorithms is crude. E.g. gradient based algorithms are not mentioned at all throughout the paper although it has been widely applied in wind energy research.

Line 76-80 The authors mention that the genetic algorithm (GA) is capable of "avoiding being trapped in local optima" and that it is used to find "the optimal solution". This sounds like global optimum is guaranteed, which is not the case for many problems. This should be made clearer.

Line 144: The authors mention that the first 6 frequencies have been investigated, but only the first 4 are shown in the table. Additionally, the mesh is also used for stress analysis. Thus, mesh convergence should be performed on stresses too, as it is a much more local phenomenon that often requires much more fine mesh resolution than natural frequencies. Natural frequencies can often be obtained accurately with a coarse mesh.

Line 179: Why do the authors alter the height to 80 m, when the mesh convergence study was made on a tower of 87.6 m?

Line 190: Perhaps figure 3 and 4 can be combined, as figure 4 also contains the geometry of the turbine tower.

Line 195-228: It is very unclear which of the formulae in section 2.3.3.1 that is applied, as it seems loads are taken directly from Lanier (2005).

Eq (1) & (2): The 50-year wind velocity, the thrust coefficient, and the rotor radius are

defined, but no values seems to be given.

Line 240-242: The authors mention that the thrust force F and bending moment My are the most significant components. This should be clarified why. Also, no coordinate system has been defined, thus My is actually not defined.

Line 245: Damage Equivalent Loads are used for fatigue damage estimation. The authors should comment on the assumptions made in the DEL method.

Line 246: The authors write that the loads from Lanier (2005) are unfactored. However, in table J-6 in Lanier (2005) both the factored and un-factored values appear. Consequently, Table 5 can be reduced.

Line 251+253: The authors refer to Lanier (2005) for both the ultimate limit loads and fatigue loads. These loads are for a hub height of 100 m, and seems to be applied directly (without any comments on this) to a tower of 80 m. This should be explained.

Line 376->: Sudden changes in geometries (thicknesses) from segment to segment will give rise to large stress concentrations,. The authors should indicate (and comment on) if the stress concentrations are taken into account or not.

Line 415->: The authors should indicate the type of buckling analysis (linear/non-linear?)

Technical comments:

Line 60 fedility -> fidelity

Line 151: force-aft -> fore-aft

Line 160: force-aft -> fore-aft

Line 240: extreme 50-year extreme wind condition -> 50-year extreme wind condition

Line 283: There seems to be a mistake in the reference listing, "Lin et. al (Wang et al.,2016) . . ."

Line 572: Allable -> allowable

---

## Referee Comment (RC2) · Anonymous Referee #2 · 6 Feb 2017

The manuscript under review presents the structural design optimization of the 5 MW NREL onshore wind turbine. Finite element technique is used to perform the evaluation of the design constrains and objective function. Genetic algorithm is use to search the design space for the optimal solution. Design variables of the tower are; top diameter, bottom diameter and thickness distribution. The design constraints are; deformation, ultimate stress, fatigue, buckling, and natural frequencies. Mass of the tower is introduced as the objective function. The optimization process resulted in 6% mass reduction of the tower, while satisfying all the design constrains.

The paper is well written in English, but it lacks several important features and details that a research paper needs to have. My recommendation is to accept the paper, but with major revision. Below are some comments for the authors to fully revise their paper:

Major comments:

- The paper lacks a critical literature review of wind turbine support structure optimization. There are several important papers in this field than need to be cited (and discussed) in the paper. Here are some examples:

- Muskulus, Michael, and Sebastian Schafhirt. "Design optimization of wind turbine support structures-a review." Journal of Ocean and Wind Energy 1, no. 1 (2014): 12-22.

- Zwick, Daniel, Michael Muskulus, and Geir Moe. "Iterative optimization approach for the design of full-height lattice towers for offshore wind turbines." Energy Procedia 24 (2012): 297-304.

- Chew, Kok-Hon, Kang Tai, E. Y. K. Ng, and Michael Muskulus. "Optimization of offshore wind turbine support structures using an analytical gradient-based method." Energy Procedia 80 (2015): 100-107.

- Schafhirt, Sebastian, Niels Verkaik, Yilmaz Salman, and Michael Muskulus. "Ultra-fast analysis of offshore wind turbine support structures using impulse based substructuring and massively parallel processors." In The Twenty-fifth International Ocean and Polar Engineering Conference. International Society of Offshore and Polar Engineers, 2015.

- Pasamontes, Lucía Bárcena, Fernando Gómez Torres, Daniel Zwick, Sebastian Schafhirt, and Michael Muskulus. "Support structure optimization for offshore wind turbines with a genetic algorithm." In ASME 2014 33rd International Conference on Ocean, Offshore and Arctic Engineering, pp. V09BT09A033-V09BT09A033. American Society of Mechanical Engineers, 2014.

- Schafhirt, Sebastian, Ana Page, Gudmund Reidar Eiksund, and Michael Muskulus. "Influence of Soil Parameters on the Fatigue Lifetime of Offshore Wind Turbines with Monopile Support Structure." Energy Procedia 94 (2016): 347-356.
- Haghi, Rad, Turaj Ashuri, Paul LC van der Valk, and David P. Molenaar. "Integrated multidisciplinary constrained optimization of offshore support structures." In Journal of Physics: Conference Series, vol. 555, no. 1, p. 012046. IOP Publishing, 2014.

- Ashuri, Turaj, Michiel B. Zaaijer, Joaquim RRA Martins, Gerard JW Van Bussel, and Gijs AM Van Kuik. "Multidisciplinary design optimization of offshore wind turbines for minimum levelized cost of energy." Renewable Energy 68 (2014): 893-905.

- Negm, Hani M., and Karam Y. Maalawi. "Structural design optimization of wind turbine towers." Computers & Structures 74, no. 6 (2000): 649-666.

- Lavassas, I., G. Nikolaidis, P. Zervas, E. Efthimiou, I. N. Doudoumis, and C. C. Baniotopoulos. "Analysis and design of the prototype of a steel 1-MW wind turbine tower." Engineering structures 25, no. 8 (2003): 1097-1106.

- Yoshida, Shigeo. "Wind turbine tower optimization method using a genetic algorithm." Wind Engineering 30, no. 6 (2006): 453-469.

- Uys, P. E., J. Farkas, K. Jarmai, and F. Van Tonder. "Optimisation of a steel tower for a wind turbine structure." Engineering structures 29, no. 7 (2007): 1337-1342.

- In the introduction section, it is not clear what the knowledge-gap is that the authors want to address in this paper. Typically, a literature review should show what the state-of-the-art is and what is still needed to be addressed. Please clarify what the contribution of this paper is to the state-of-the-art, based on the identified knowledge gap.

- Line 179: Why did the authors change the tower height to 80 m? Mesh sensitivity has a different height. Please explain.

- Line 204: Gravity is a body load, but the authors applied it to the tower top as a nodal load. Please explain why such a choice is made, and how accurate such an assumption will be. If gravity is used as a force, what is the corresponding mass? The authors could apply gravity as an acceleration in their model, and use an element type

that would consider gravitational acceleration.

- What values are used for equation 1 (and 2, 3 ...)? Please add them in the paper.

- What approach is used to do the fatigue analysis? What properties are used to find the damage on the structure? Please elaborate how the fatigue damage is obtained from iteration to iteration? Is there any time domain analysis performed? Is the Miner rule or Paris formulas used? Section 2.3.3.2. needs more details to allow other authors replicate the same analysis if needed.

- Line 262, "For ultimate load case, both gravity loads due to the weight of the tower itself and the wind loads due to wind passing the tower are taken into account as distributed loads on the tower ...", is there any time-domain simulation used here or is this just a static analysis? How is the tower shadow considered in this work? Please clarify.

- Section 3: GA is a standard and routine approach and no need to spend 2 pages on that. Few citations to relevant paper would do the job. Please consider removing it and adding few citations instead.

- Section 4.2: Why does the tower geometry vary linearly? Is this done to save computational time? Such an assumption has increased the initial weight of the tower. Please explain by how much?

- Line 374: Such an assumption is not acceptable. To do an optimization of the 5 MW NREL tower, the authors should use the same deformation of the tower top as the original design. You can relax a constraints to always have a better design in an optimization study. This is a very crude approach used in the paper.

- Line 395: This is not an acceptable assumption. How do you know that the tower oscillates at the same RPM as the rotor to find the number cycles? Please clarify.

- Page 14, Fatigue constraint: How did you compute the fatigue damage? How did the authors find the cumulative stresses to be used in the S-N curve?

- Line 415: Please elaborate how the buckling is performed, and how is the design guard against buckling?

- Line 440: The rotor rotational speed is 12.1 RPM and not 11.2 RPM. I hope this to be a typo, otherwise the optimization has to be redone.

- I do not see an evaluation of the fatigue damage in the results section. Please add this, since it is important.

- Section 4.5: I expect this to be in the methodology section, particularly figure 7. Please consider making a new section named methodology to explain how you setup your research, and how the design constraints, objective function and optimization algorithm are defined.

- There is no soil-structure interaction considered. Please explain why and how you would this impact your results.

- How is the mesh perturbation done in this work? Please explain.

Minor comments:

- In table 1, please also add the Poisson's ratio.

- Line 175, "80m", add half-space.

- All the equations seem to be copied from somewhere instead of being typeset. Please consider typesetting them.

- Math symbol sub-indexes in Table 5, 6 ... are difficult to read. Please increase the font size.

- Table 8 to be left adjusted.

- Correct "force-aft" in line 151 and 160 to "fore-aft".

---

## Author Comment (AC1) · 5 Mar 2017

Dr Lin Wang
Research Fellow
Offshore Renewable Energy Centre
School of Water, Energy and Environment
Cranfield University
Cranfield, Bedford, MK43 0AL
UK

Dear Reviewer,

We appreciate very much for your comments. We were asked to response to all comments, while a revised manuscript should not be prepared at this stage. In the following, we will therefore engage with all the comments and propose improvements for the final manuscript.

1) The introduction is lacking a more thorough overview of structural optimisation of wind turbine towers/support structures, which is the main aspect of the paper. It seems that mainly optimization using GA is listed, and these references appear in a later section (Section 3). Also, most references in the paper are to blade design rather than tower design.

Our response:
A more thorough overview of structural optimisation of wind turbine towers/support structures will be added in the revised paper. More papers related to the tower design will be discussed and cited.

2) Line 64-65 The categorization of optimization algorithms is crude. E.g. gradient based algorithms are not mentioned at all throughout the paper although it has been widely applied in wind energy research.

Our response:
The categorization of optimization algorithms will be revised, covering gradient based algorithms.

3) Line 78-80 The authors mention that the genetic algorithm (GA) is capable of "avoiding being trapped in local optima" and that it is used to find "the optimal solution". This sounds like global optimum is guaranteed, which is not the case for many problems. This should be made clearer.

Our response:
The statement "avoiding being trapped in local optima" will be removed, and a more precise description of capability of GA will be added in the revised paper.

4) Line 144: The authors mention that the first 6 frequencies have been investigated, but only the first 4 are shown in the table. Additionally, the mesh is also used for stress analysis. Thus, mesh convergence should be performed on stresses too, as it is a much more local phenomenon that often requires much more fine mesh resolution than natural frequencies. Natural frequencies can often be

obtained accurately with a coarse mesh.

Our response:
The "first 6 frequencies" was a typo. It should be "first 4 frequencies". The typo will be corrected in the revised paper.

Mesh convergence on stresses will be performed in the revised paper.

5) Line 179: Why do the authors alter the height to 80m, when the mesh convergence study was made on a tower of 87.6m?

Our response:
The optimisation framework developed in this work is generic in nature and can be applied to the structural optimisation of wind turbine towers with an arbitrary height. In this paper, the NREL 5MW wind turbine 87.6m-height tower is used only for validation purpose. For the optimisation case study, a typical value of 80m is chosen as the height of the tower.

6) Line 190: Perhaps figure 3 and 4 can be combined, as figure 4 also contains the geometry of the turbine tower.

Our response:
Figs. 3 and 4 will be combined in the revised paper.

7) Line 195-228: It is very unclear which of the formula in Section 2.3.3.1 that is applied, as it seems loads are taken directly from Lanier (2005). Eq. (1) & (2): The 50-year wind velocity, the thrust coefficient, and the rotor radius are defined, but no values seems to be given.

Our response:
The aerodynamic loads on the rotor, as listed in Tables 5 and 6, are taken directly from Lanier (2005). Eq. (1) in Section 2.3.3.1 was added to present the formula which can be used to calculate the aerodynamic thrust force on the parked rotor. Eq. (1) will be removed in the revised paper, and statements will be added to clearly indicate the aerodynamic loads on the rotor are taken directly from Lanier (2005).

Eqs. (2) and (3) in Section 2.3.3.1 were used in this paper to calculate the wind loads on the tower itself.

8) Line 240-242: The authors mention that the thrust force F and bending moment My are the most significant components. The should be clarified why. Also, no coordinate system has been defined, thus My is actually not defined.

Our response:
The thrust force F and bending moment My are generally considered as the most significant

components in aerodynamic loads. References will be added to support this statement. Coordinate system will be defined in the revised paper to facilitate the definition of force F and bending moment My.

9) Line 245: Damage Equivalent Loads are used for fatigue damage estimation. The authors should comment on the assumptions made in the DEL method.

Our response:
A discussion on the DEL method will be added in the revised paper.

10) Line 246: The authors write that the loads from Lanier (2005) are unfactored. However, in Table J-6 in Lanier (2005) both the factored and unfactored values appear. Consequently, Table 5 can be reduced.

Our response:
Table 5 will be reduced to only present the factored values.

11) Line 251+253: The authors refer to Lanier (2005) for both the ultimate limit loads and fatigue loads. These loads are for a hub height of 100m, and seems to be applied directly (without any comments on this) to a tower of 80m. This should be explained.

Our response:
Detailed rotor aerodynamic load calculations, which are generally based on BEM or CFD, is out of the scope of the paper. Therefore, the loads from Lanier (2005) are used in this study as representative rotor aerodynamic loads for 5MW wind turbines, which may be placed at different tower heights (e.g. 80m, 90m, 100m etc.). This will be clearly indicated in the revised paper.

12) Line 276->: Sudden changes in geometries (thicknesses) from segment to segment will give rise to large stress concentrations. The authors should indicate (and comment on) if the stress concentrations are taken into account or not.

Our response:
Discussion on stress concentrations will be added in the revised paper.

13) Line 415->: The authors should indicate the type of buckling analysis (linear/nonlinear)

Our response:
The buckling analysis performed in this paper is linear. This will be indicated clearly in the revised paper.

14) Technical comments:
Line 60 fedility -> fidelity
Line 151: force-aft -> fore-aft
Line 160: force-aft -> fore-aft

Line 240: extreme 50-year extreme wind condition -> 50-year extreme wind condition
Line 283: There seems to be a mistake in the reference listing, "Lin et. Al (Wang et al., 2016)…"

Our response:
These typos will be corrected in the revised paper.

Best regards,
Lin

---

## Author Comment (AC2) · 5 Mar 2017

Dr Lin Wang
Research Fellow
Offshore Renewable Energy Centre
School of Water, Energy and Environment
Cranfield University
Cranfield, Bedford, MK43 0AL
UK

Dear Reviewer,

We appreciate very much for your comments. We were asked to response to all comments, while a revised manuscript should not be prepared at this stage. In the following, we will therefore engage with all the comments and propose improvements for the final manuscript.

1) The paper lacks a critical literature review of wind turbine support structure optimization. There are several important papers in this field than need to be cited (and discussed) in the paper. Here are some examples:
- Muskulus, Michael, and Sebastian Schafhirt. "Design optimization of wind turbine support structures – a review." Journal of Ocean and Wind Energy 1, no. 1 (2014): 12-22.
- Zwick, Daniel, Michael Muskulus, and Geir Moe. "Iterative optimization approach for the design of full-height lattice towers for offshore wind turbines." Energy Procedia 24 (2012):297-304.
- Chew, Kok-Hon, Kang Tai, E. Y. K. Ng, and Michael Muskulus. "Optimization of offshore wind turbine support structures using an analytical gradient-based method". Energy Procedia 80 (2015): 100-107.
- Schafhirt, Sebastian, Niels Verkaik, Yilmaz Salman, and Michael Muskulus. "Ultrafast analysis of offshore wind turbine support structures using impulse based substructuring and massively parallel processors." In the Twenty-fifth International Ocean and Polar Engineering Conference. International Society of Offshore and Polar Engineers. 2015.
- Pasamontes, Lucia Barcena, Fernando Gomez Torres, Daniel Zwick, Sebastian Schafhit, and Michael Muskulus. "Support structure optimization for offshore wind turbines with genetic algorithm." In ASME 2014 33rd International Conference on Ocean, Offshore and Arctic Engineering, pp. V09BT09A033-V09BT09A033. American Society of Mechanical Engineers, 2014.
- Schafhirt, Sebastian, Ana Page, Gudmund Reidar Eiksund, and Michael Muskulus. "Influence of Soil Parameters on the Fatigue Lifetime of Offshore Wind Turbines with Monopile Support Structure". Energy Procedia 94 (2016): 347-356.
- Haghi, Rad, Turaj Ashuri, Paul LC van der Valk, and David P. Molenaar. "Integrated multidisciplinary constrained optimization of offshore support structures." In Journal of Physics: Conference Series, vol. 555, no. 1, p. 012046. IOP Publishing, 2014.
- Ashuri, Turaj, Michiel B. Zaaijer, Joaquim RRA Martins,, Gerard JW Van Bussel, and Gijs AM Van Kuik. "Multidisciplinary design optimization of offshore wind turbines for minimum levelized cost of

energy." Renewable Energy 68 (2014): 893-905.

- Negm, Hani M., and Karam Y. Maalawi. "Structural design optimization of wind turbine towers." Computers & Structures 74, no. 6 (2000): 649-666.

- Lavassas, I., G. Nikolaidis, P. Zervas, E. Efthimiou, I.N. Doudoumis, and C.C. Baniotopoulos. "Analysis and design of the prototype of a steel 1-MW wind turbine tower." Engineering structures 25, no. 8 (2003): 1097-1106.

- Yoshida, Shigeo. "Wind turbine tower optimization method using a genetic algorithm." Wind Engineering 30, no. 6 (2006): 453-469.

- Uys, P.E., J. Farkas, K. Jarmai, and F. Van Tonder. "Optimisation of steel tower for a wind turbine structure." Engineering structures 29, no. 7 (2007): 1337-1342.

Our response:
A critical literature review of wind turbine tower and support structure optimization will be added, citing and discussing the important papers in this field.

2) In the introduction section, it is not clear what the knowledge-gap is that the authors want to address in this paper. Typically, a literature review should show what the state-of-the-art is and what is still needed to be addressed. Please clarify what the contribution of this paper is to the state-of-the-art, based on the identified knowledge gap.

Our response:
The knowledge-gap and the contribution of this paper will be added in the introduction section.

3) Line 179: Why did the authors change the tower height to 80 m? Mesh sensitivity has a different height. Please explain.

Our response:
The optimisation framework developed in this work is generic in nature and can be applied to the structural optimisation of wind turbine towers with an arbitrary height. In this paper, the NREL 5MW wind turbine 87.6m-height tower is used only for validation purpose. For the optimisation case study, a typical value of 80m is chosen as the height of the tower.

4) Line 204: Gravity is a body load, but the authors applied it to the tower top as a nodal load. Please explain why such a choice is made, and how accurate such an assumption will be. If gravity is used as a force, what is the corresponding mass? The authors could apply gravity as an acceleration in their model, and use an element type that would consider graviational acceleration.

Our response:
The gravity loads due to mass of the components on the tower top (such as the rotor and nacelle) are taken into account by applying a point mass with a value of 480,076 (LaNier, 2005) on the tower top. The gravity loads due to the mass of the tower itself are taken by ANSYS software automatically through defining gravity acceleration. This will be indicated clearly in the revised paper.

5) What values are used for equation 1 (and 2, 3 …)? Please add them in the paper.

Our response:
The aerodynamic loads on the rotor, as listed in Tables 5 and 6, are taken directly from Lanier (2005). Eq. (1) in Section 2.3.3.1 was added to present the formula which can be used to calculate the aerodynamic thrust force on the parked rotor. Eq. (1) will be removed in the revised paper, and statements will be added to clearly indicate the aerodynamic loads on the rotor are taken directly from Lanier (2005).

Eqs. (2) and (3) in Section 2.3.3.1 were used in this paper to calculate the wind loads on the tower itself. Values used in Eqs. (2) and (3) will be added in the revised paper.

6) What approach is used to do the fatigue analysis? What properties are used to find the damage on the structure? Please elaborate how the fatigue damage is obtained from iteration to iteration? Is there any time domain analysis performed? Is the Miner rule or Paris formulas used? Section 2.3.3.2 needs more details to allow other authors replicate the same analysis if needed.

Our response:
The S-N curve method is used in the fatigue analysis, as presented in Fatigue Constraint in Section 4.3. More details about the fatigue analysis will be added in the revised paper.

7) Line 262, "For ultimate load case, both gravity loads due to the weight of the tower itself and the wind loads due to wind passing the tower are taken into account as distributed loads on the tower ..", is there any time-domain simulation used here or is this just a static analysis? How the tower shadow considered in this work? Please clarify.

Our response:
It is a static analysis. The tower shadow is not considered in this work, as the tower shadow effects are deemed negligible in the case studies performed in this work. This will be clearly indicated in the revised paper.

8) Section 3: GA is a standard and routine approach and no need to spend 2 pages on that. Few citations to relevant paper would do the job. Please consider removing it and adding few citations instead.

Our response:
Section 3 will be revised to make it more concise.

9) Section 4.2: Why does the tower geometry vary linearly? Is this done to save computational time? Such an assumption has increase the initial weight of the tower. Please explain by how much?

Our response:
It is the common practice to design the tower with a linear variation of outer diameters across the

length of the tower. Examples can be seen in NREL 5MW wind turbine tower and DTU 10MW wind turbine tower [1]. These will be added in the revised paper to support the assumption of linearly varied tower outer diameters.

[1] Christian Bak, et al. "Description of the DTU 10 MW Reference Wind Turbine", DTU Wind Energy Report-I-0092, 2013

10) Line 374: Such an assumption is not acceptable. To do an optimization of the 5 MW NREL tower, the authors should use the same deformation of the tower top as the original design. You can relax a constraint to always have a better design in an optimization study. This is a very crude approach used in the paper.

Our response:
In this paper, the NREL 5MW wind turbine 87.6m-height tower is used only for validation purpose. For the optimisation case study, a 80m-height tower is considered. This will be clearly indicated in the revised paper.

11) Line 395: This is not an acceptable assumption. How do you know that the tower oscillates at the same RPM as the rotor to find the number cycles? Please clarify.

Our response:
Due to the rotor rotation and wind shear, the rotor aerodynamic thrust force Fx and bending moment My transferable to the tower top are cyclic loads, of which frequency is associated with the frequency of rotor rotation. This assumption is then used to estimate the number of cycles. Clarification on this assumption will be added in the revised paper.

12) Page 14: Fatigue constraint: How did you compute the fatigue damage? How did the authors find the cumulative stresses to be used in the S-N curve?

Our response:
It should be noted that the fatigue loads (see Table 6) taken from LaNier (2005) are load range. With load range, the stress range, which is used in the S-N curve, is determined using the parametric FEA model.

13) Line 415: Please elaborate how the buckling is performed, and how is the design guard against buckling?

Our response:
The details of buckling analysis will be added in the revised paper.

14) Line 440: The rotor rotational speed is 12.1 RPM and not 11.2 RPM. I hope this to be a typo, otherwise the optimization has to be redone.

Our response:

It is a typo and will be corrected in the revised paper.

15) I do not see an evaluation of the fatigue damage in the results section. Please add this, since it is important.

Our response:

The evaluation of the fatigue damage will be added in the results section in the revised paper.

16) I expect this to be in the methodology section, particularly figure 7. Please consider making a new section named methodology to explain how you setup your research, and how the design constraints, objective function and optimization algorithm are defined.

Our response:

Fig. 7 is in the Section 4.4. Section 4 is the methodology section, covering objective function (Section 4.1), design variables (Section 4.2), constraints, parameter settings of genetic algorithm (Section 4.3), and flowchart of the optimisation model (Section 4.4).

17) There is no soil-structure interaction considered. Please explain why and how you would this impact your results.

Our response:

The soil-structure interaction is important for the design of offshore wind turbine support structures. However, as indicated in the end of the introductory part of the paper, this paper deals with onshore wind turbine towers, of which design usually does not consider soil-structure interactions.

18) How is the mesh perturbation done in this work? Please explain.

Our response:

The element size, which is determined from mesh convergence exercises, is fixed during the optimisation process.

19) Minor comments:
- In table 1, please also add the Poisson's ratio.
- Line 175, "80m", add half-space.
- All the equations seem to be copied from somewhere instead of being typeset. Please consider typesetting them.
- Math symbol sub-indexes in Table 5,6 … are difficult to read. Please increase the font size.
- Table 8 to be left adjusted.
- Correct "force-aft" in line 151 and 160 to "fore-aft".

Our response:

Typos will be corrected and the format will be adjusted in the revised paper.

Best regards,
Lin